# Systematic Assessment of Factual Knowledge in Large Language Models

**Linhao Luo  Thuy-Trang Vu  Dinh Phung  Gholamreza Haffari**
Department of Data Science and AI
Faculty of Information Technology, Monash University, Australia
`{linhao.luo,trang.vu1,dinh.phung,gholamreza.haffari}@monash.edu`

## Abstract

Previous studies have relied on existing question-answering benchmarks to evaluate the knowledge stored in large language models (LLMs). However, this approach has limitations regarding factual knowledge coverage, as it mostly focuses on generic domains which may overlap with the pretraining data. This paper proposes a framework to systematically assess the factual knowledge of LLMs by leveraging knowledge graphs (KGs). Our framework automatically generates a set of questions and expected answers from the facts stored in a given KG, and then evaluates the accuracy of LLMs in answering these questions. We systematically evaluate the state-of-the-art LLMs with KGs in generic and specific domains. The experiment shows that ChatGPT is consistently the top performer across all domains. We also find that LLMs performance depends on the instruction finetuning, domain and question complexity and is prone to adversarial context.[1]

## 1 Introduction

The rise of Large Language Models (LLMs) has greatly improved the capabilities of natural language processing (NLP). However, one primary concern with these models is the potential for *extrinsic hallucinations* where LLMs generate statements that cannot be verified from the source (Levy et al., 2021; Ji et al., 2023). This issue severely impairs the trustworthiness of LLMs and is particularly concerning when relying on LLMs for decision-making. Rigorous evaluation is necessary before deploying them in critical applications.

One evaluation approach is to use question-answering datasets to assess the language and knowledge capabilities of LLMs. Recent research has mainly focused on evaluation using existing benchmarks (Bommasani et al., 2023; Bang et al.,

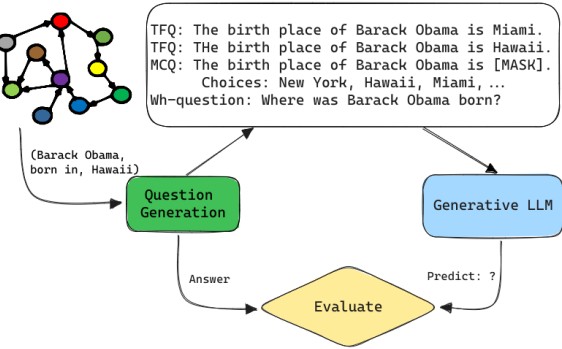

Figure 1: Our proposed assessment framework generates a diverse set of questions to evaluate factual knowledge in LLMs.

2023; Guo et al., 2023). While these benchmarks are valuable for comparison and measuring progress in LLM research, they may not provide sufficient assessment for production. Benchmarks constructed from public datasets can pose information leakage problems due to overlap with pretraining data. Furthermore, constructing domain-specific benchmarks is costly, requiring domain expertise and adequate knowledge coverage.

This paper proposes a systematic approach to assess factual knowledge in LLMs by generating a comprehensive assessment suite from knowledge graphs (KGs) and evaluating the correctness of LLMs' responses. The question generation process is carefully designed to ensure coverage of facts, as well as diversity and validity of the questions (Figure 1). Using this framework, we evaluate multiple models from three LLM families on factual questions derived from four KGs, covering both generic and specialized domains. Specifically, our contributions are:

- We propose a novel framework to evaluate factual knowledge in LLMs by systematically generating valid and diverse questions from KGs while also ensuring knowledge coverage.

- We observe that LLMs may abstain from an-

---

[1]Code and data will be released at `https://github.com/RManLuo/llm-facteval`

swering certain questions, prioritizing precision by avoiding the provision of inaccurate or hallucinated answers. We propose to use the F1 metric to take the abstention into account and ensure fair comparison across models.

- We show that LLMs performance depends on several factors such as instruction finetuning, domains, and question complexity. Despite sharing the same parametric knowledge base, models finetuned with different instruction datasets show varying performance levels. In general-domain KGs, LLMs achieve the highest score, but their performance declines in specialized domains and is worse on questions having a wide range of potential answers.

- We assess robustness of LLMs to the prompting context and find they are highly sensitive to irrelevant information and are susceptible to being misled by antifactual contexts.

## 2 Systematic Assessment Framework

This section describes the question generation component in our proposed assessment framework, followed by the answer prompting strategy to collect LLM's response and the evaluation metric.

### 2.1 Question Generation

Our framework leverages the facts stored in a KG, organized into triplets, i.e., *(subject, relation label, object)*, to automatically generate a set of knowledge-based questions and answers satisfying three requirements: (i) *validity*: questions should have *unique* or *verifiable* answers ; (ii) *coverage*: questions should cover all explicit facts; and (iii) *diversity*: questions should vary in format and difficulty.

In this paper, we assume the complete KG and generate valid questions by considering the object of a given triplet as the reference answer and generating questions with the subject and relation label. To ensure the question coverage and diversity, we utilize all available triplets and employ two question generation methods from a predefined template (Petroni et al., 2019) or using ChatGPT (OpenAI, 2023). We consider three *types* of questions: true-false question (TFQ), multiple choice question (MCQ), and short-answer question (SAQ). In addition, each question type can be represented in different *formats*: true/false question, fill-in-the-bank (FiB) question, and Wh- question (Figure 3

in Appendix).

**True-false question (TFQ)** Given a triplet, we create factual questions that ask the LLM to determine whether a given statement is true or false. For example, given the triplet *(Barack Obama, born in, Hawaii)*, we can generate a true statement *"The birth place of Barack Obama is Hawaii."*. For false statement, we randomly replace the object with a wrong entity.

**Multiple choice questions (MCQ)** The LLM is presented with a list of answer candidates (choices) and is required to select the correct one. The candidates consist of the object along with randomly selected incorrect entities. We consider two formats for MCQ: fill-in-the-blank (FiB) by replacing the reference object in the true statement with [MASK] token and Wh-question (Aigo et al., 2021).

**Short-answer questions (SAQ)** Instead of providing answer candidates as in MCQ, we ask the LLM to predict the correct answer directly in SAQ. For many-to-many relations, we consider all possible objects as potential correct answers and request the LLMs to list all possible answers.

### 2.2 Evaluation

**Answer Prompting** We carefully design prompts to describe the task and instruct the LLMs to provide concise answers. We also verify the robustness and consistency of LLMs by injecting different types of knowledge into the question, including (i) *relevant* knowledge, (ii) *irrelevant* knowledge which is correct but not related to the question, and (iii) *anti-factual* knowledge that provides false or erroneous information. The injected knowledge can come from the relation description or extra evidence information, which are available in several knowledge graphs.

**Metric** Although we prompt LLMs to provide brief and concise answers, evaluating the correctness of the generated answer is not trivial. A small percentage of generated answers are long and contain explanations. Hence, the standard exact match metric used in question-answering tasks is not a suitable metric. Instead, we use a fuzzy match metric that checks if the generated answer appears in the reference answers and vice versa.

Many LLMs employ several guardrails to avoid providing inaccurate or hallucinated answers which return an abstained answer (e.g., "I am unable to answer the questions without more knowledge."). We

| Model | TFQ | | MCQ | | | SAQ | | | AVG |
|---|---|---|---|---|---|---|---|---|---|
| | TPL | GPT3.5 | FiB-TPL | FiB-GPT3.5 | Wh-GPT3.5 | FiB-TPL | FiB-GPT3.5 | Wh-GPT3.5 | |
| ChatGPT | **76.98** | 77.43 | **60.94** | **63.82** | 50.63 | 5.13 | 2.47 | **19.57** | **44.62** |
| LLaMA-7B | 1.23 | 1.46 | 7.20 | 0.76 | 0.27 | 2.93 | 3.07 | 0.15 | 2.13 |
| Alpaca | 65.07 | 60.65 | 41.95 | 40.50 | 41.68 | 7.68 | 6.01 | 8.42 | 34.00 |
| Vicuna | 52.83 | 51.84 | 15.47 | 18.28 | 33.84 | 3.79 | 4.81 | 7.83 | 23.59 |
| T5-XL | 23.87 | 6.79 | 5.77 | 3.96 | 8.23 | 1.63 | 1.69 | 1.33 | 6.66 |
| FLAN-T5-XL | 75.01 | **79.75** | 51.59 | 50.72 | **51.66** | **11.57** | **11.42** | 9.99 | 42.71 |
| FLAN-Alpaca | 54.83 | 53.08 | 46.59 | 47.58 | 48.59 | 9.89 | 8.06 | 10.93 | 34.94 |
| FLAN-Vicuna | 63.73 | 63.80 | 46.80 | 46.76 | 48.69 | 2.74 | 2.89 | 10.83 | 35.78 |

| % abstention | 0% | 25% | 50% | 75% | 100% |
|---|---|---|---|---|---|

Table 1: Precision of LLMs on different types of questions generated from Google-RE: true-false question (TFQ), multi-choice question (MCQ) and short-answer questions (SAQ). SAQ and MCQ questions can be in fill-in-the-blank (FiB) or Wh-question (Wh) format. TPL and GPT3.5 denote whether the questions are generated by the template and GPT3.5, respectively. The shade of the background color shows the percentage of abstained responses. The best performance of each question type is marked in **bold**.

define precision as the accuracy of non-abstained answers

$$P = \frac{correct}{correct + incorrect} \quad (1)$$
$$P^* = P \times (1 - A) \quad (2)$$

and recall as the percentage of accuracy of all questions

$$R = \frac{correct}{correct + incorrect + abstained} \quad (3)$$

The F1 score $F1 = 2 \times \frac{P \times R}{P+R}$ is the main evaluation metric to compare the performance of LLMs.

## 3 Experiments

### 3.1 Setup

**Datasets** We use four KGs in LAMA (Petroni et al., 2019) and BioLAMA (Sung et al., 2021) benchmarks to generate factual questions, including two general-domain KGs: Google-RE (Petroni et al., 2019), T-REx (Elsahar et al., 2018), and two domain-specific KGs: WikiBio (Sung et al., 2021) in biology domain and ULMS (Bodenreider, 2004) in the medical domain. Each relation in the KGs is associated with a predefined template to construct a natural sentence from a given triplet. Detail descriptions of the datasets and the predefined templates are reported in Appendix A.1.

**Large Language Models** We evaluate the knowledge captured in several LLMs coming from three backbones: (i) ChatGPT[2] (OpenAI, 2023);

---

[2]We did not assess GPT4 due to its high cost.

(ii) LLaMA family, including LLaMA-7B (Touvron et al., 2023), Alpaca (Taori et al., 2023) and Vicuna (Chiang et al., 2023) which are instruction finetuned from LLaMA-7B backbone; and (iii) T5 family, including T5-XL (Raffel et al., 2020), FLAN-T5 XL (Chung et al., 2022) and two FLAN-T5-XL-based models which are instruction finetuned on Alpaca and Vicuna datasets, denoted as FLAN-Alpaca (Chia et al., 2023) and FLAN-Vicuna respectively. The details regarding prompting can be found in Appendix A.2.

**Experiment Settings** We employ two question generation methods: (i) template-based (TPL) where the subject is plugged into the provided template and the object is the ground-truth answer; and (ii) LLM-based where we use GPT-3.5-turbo to generate the questions. The question generation prompt can be found in Appendix C. Given a triplet, we generate the TFQ with the ratio of true and false questions set to 1 : 1. For MCQ, we randomly select three incorrect entities and combine them with the correct entities as the choices.

### 3.2 Results

**Precision** We report the precision of LLMs on question generated from Google-RE in Table 1. As expected, LLMs perform best on TFQ and worst on SAQ due to the increasing difficulty level. Surprisingly, almost all LLMs struggle with FiB questions, often returning abstentions or the [MASK] token without making any predictions. While FiB questions are commonly used in masked language model evaluation, we find that Wh-questions, which are more natural and occur more frequently

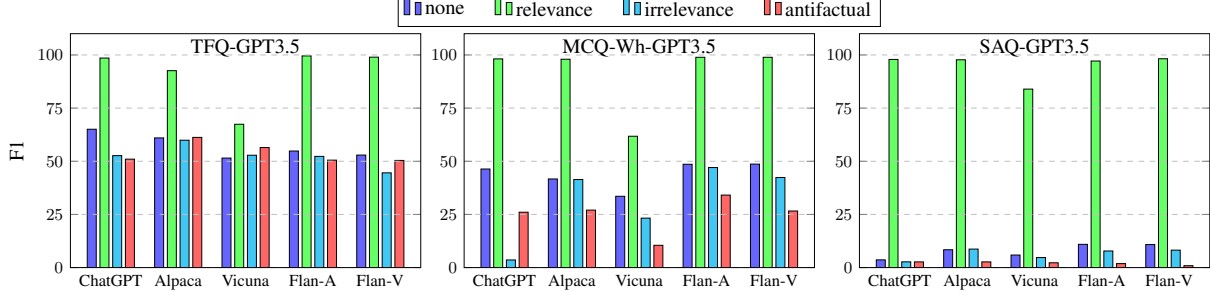

Figure 2: F1 score of LLMs on Google-RE with different context prompt: none, relevant, irrelevance and antifactual context (best seen in color). FLAN-A and FLAN-B denote `FLAN-Alpaca` and `FLAN-Vicuna` respectively.

in the instruction set, are more suitable for evaluating conversational LLMs. Moreover, we observe comparable performance between template-based and GPT3.5-based questions.

Overall, `ChatGPT` achieves the best average precision. However, it also has a high percentage of abstained answers across all question types. Both the `LLaMA-7B` and `T5-XL` models perform worse than random guessing in TFQ and MCQ, indicating a failure to follow instructions due to the lack of training on instruction finetuning datasets. Although sharing the same parametric knowledge base (`LLaMA-7B`), `Alpaca` consistently outperforms `Vicuna`. On the other hand, further instruction fine-tuning the `FLAN-T5-XL` does not improve precision.

**F1 Measure** Table 2 shows the average F1 score across all question types for each KG. The detailed breakdown of F1 scores for each question type can be found in Appendix B. Overall, `ChatGPT` outperforms other LLMs, and models from the T5 family generally perform better than those from the LLaMA family. Among the models from the same family, those fine-tuned on the `Alpaca` instruction set have better performance. This contrasts with the above observation where `FLAN-T5-XL` is the top performer in terms of precision in the T5 family. With the high abstention rate, it can be seen that `FLAN-T5-XL` tends to abstain from uncertain questions to achieve higher precision, which comes at the expense of losing recall for correct answers.

**Impact of Domain** As shown in Table 2, the F1 scores on TREx (general domain) are higher than those in specific domains (WikiBio and UMLS). Additionally, the relatively stronger performance on WikiBio over UMLS can be attributed to the pretraining data overlap as it is derived from Wikipedia. Interestingly, all LLMs perform poorly on the Google-RE dataset, despite also being extracted from the general domain (Wikipedia). We

| | Google-RE | TREx | WikiBio | UMLS |
|---|---|---|---|---|
| ChatGPT | **35.77** | **74.00** | **62.74** | **48.99** |
| LLaMA-7B | 2.07 | 8.49 | 1.26 | 1.35 |
| Alpaca | 32.56 | 61.00 | 41.99 | 36.84 |
| Vicuna | 22.86 | 41.08 | 25.78 | 22.99 |
| T5-XL | 6.65 | 11.31 | 9.51 | 15.35 |
| FLAN-T5-XL | 30.62 | 57.14 | 35.82 | 30.33 |
| FLAN-Alpaca | 34.89 | 58.41 | 36.13 | 35.39 |
| FLAN-Vicuna | 32.69 | 54.60 | 36.60 | 34.91 |

Table 2: Average F1 score of LLMs across question types on different KGs. The best score is in **bold**.

speculate that this discrepancy may be attributed to the complexity of the answer range of the Google-RE questions such as date-of-birth, birth place, and death place which have a wide answer range.

**Robustness to Adversarial Context** We inject different contexts to the questions of Google-RE evaluation set and reported the results in Figure 2. Our observations reveal that the responses of LLMs are highly sensitive to the contexts. Incorporating relevant context leads to significant performance improvement across all LLMs. Conversely, LLMs are prone to be misled by antifactual context, despite explicitly instructed to base their answers on real-world facts. LLMs performance also decrease when conditioned on irrelevant contexts. These findings highlight the lack of robustness in LLMs against adversarial examples. Ideally, a robust LLM should perform comparable in the absence of context or with irrelevant context. This poses a challenge in deploying LLMs to production, as they may inadvertently reinforce misinformation provided by users.

## 4  Related Works

**LLM Evaluation** Evaluation of the Large Language Model (LLM) has gained increasing interest among researchers (Bommasani et al., 2023; Bang

et al., 2023; Guo et al., 2023). For instance, Bang et al. (2023) conducts a multitask, multilingual, and multimodal evaluation for ChatGPT. Holistic Evaluation of Language Models (HELM) (Bommasani et al., 2023) selects a broad of datasets and benchmarks to evaluate the ability of LLMs. However, previous works mostly focus on human evaluation and using existing datasets and benchmarks (Guo et al., 2023). This requires lots of human effort and cannot guarantee the knowledge coverage to assess knowledge in LLMs comprehensively.

**Factual Knowledge Evaluation for LLMs**   Evaluating the factual knowledge of LLMs can ensure the model is providing reliable and trustworthy information to users. Knowledge Graphs (KGs), which capture vast amounts of facts, offer a reliable source of factual knowledge for evaluation (Pan et al., 2023; Luo et al., 2023). LAMA (Petroni et al., 2019) adopts pre-defined templates to convert the facts in KGs into cloze questions then uses LLMs to predict the answers. The prediction results are used to evaluate the knowledge stored in LLMs. Similarly, BioLAMA (Sung et al., 2021) and MedLAMA (Meng et al., 2021) assess the factual knowledge of LLMs in medical domains by using medical knowledge graphs. Alex et al. (Mallen et al., 2022) selects unpopular facts from Wikidata knowledge graphs which have low-frequency clicked entities to investigate the ability of LLMs to retain less popular factual knowledge. By enumerating all available factual triplets in KGs, we could ensure the evaluation coverage of the factual knowledge. Nevertheless, exciting methods lack a systematic framework containing question generation and evaluation modules. They often use pre-defined templates for question generation which cannot provide diverse questions to evaluate the knowledge of instruction-tuning LLMs (Sun et al., 2023).

**Automatically Question Generation from KGs**
To assess knowledge in instruction-tuning LLMs, we need to evaluate whether they have such knowledge and whether they can accurately express their knowledge, i.e. instruct following ability and robustness. Therefore, given the same factual knowledge, we need to generate diverse questions at different levels of difficulty. Early works that generate questions from KGs either use sequence-to-sequence models or graph neural networks to convert the triplet into a natural language question (Seyler et al., 2017; Kumar et al., 2019; Indurthi et al., 2017; Chen et al., 2023). Recently, many methods harness the ability of LLMs to generate questions from KGs (Guo et al., 2022; Axelsson and Skantze, 2023). In this way, they can generate questions with different diversities and complexities. Although there are previous works that generate questions from knowledge graphs, to the best of our knowledge, none of them adopt the generated questions for evaluating the factual knowledge in LLMs.

## 5   Conclusion

We propose a systematic framework to evaluate factual knowledge of LLMs with the diverse and well-coverage questions generated from KG. The experiment reveals several factors affecting LLMs' performance and highlights their vulnerability to adversarial context. Our findings contribute to understanding LLMs' capabilities and limitation in handling factual knowledge.

## Limitations

The limitation of our work includes

- Assuming a completed knowledge graph. In our work, we access the knowledge of LLMs by using the facts in knowledge graphs. However, knowledge graphs are often incomplete, which could contain lots of implicit facts. Thus, it could be inadequate to evaluate the LLMs with the existing KGs. In the future, we plan to incorporate the knowledge graph completion methods and present a more comprehensive assessment framework.

- Focusing only on triplet-based facts. We only assess the knowledge of LLMs by using the question generated from the single triplet, which ignores the complex knowledge represented by the combination of triplets. To assess the completed knowledge, we need to design a framework that considers the reasoning ability of LLMs on knowledge graphs.

- Evaluating the correctness of multiple answer questions. For N-M relations, we have multiple answers to a question. However, the LLMs might not return all the answers. How to evaluate the partially answered questions is still an open question for accessing the knowledge of LLMs.

## Ethics Statement

Our work aims to design a framework that can automatically assess the factual knowledge stored in large language models. In this research, we conducted experiments on publicly available datasets and implemented our approaches using commonly accepted techniques, giving utmost consideration to fairness and avoiding potential biases. We acknowledge the significance of transparency and have furnished comprehensive elucidations regarding our methodology and decision-making process. To conclude, our research adheres to ethical guidelines and poses no potential risks.

## Acknowledgments

This research is supported by the ARC Future Fellowship FT190100039. The authors are grateful to the anonymous reviewers for their helpful comments.

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

| KGs | Domain | #Relations | #Entities | #Triplet |
|-----|--------|-----------:|----------:|---------:|
| Google-RE | General | 3 | 7,242 | 55,28 |
| T-Rex | General | 46 | 31,180 | 34,039 |
| WikiBio | Biology | 5 | 68,39 | 17,582 |
| ULMS | Medical | 17 | 18,910 | 64,305 |

Table 3: Dataset Statistics

# A  Implementation and Experiment Detail

## A.1  Dataset

We evaluate LLMs with the knowledge derived from the following KGs

- T-REx (Elsahar et al., 2018), a knowledge graph extracted from Wikipedia. T-REx includes a relation label, a description, and a template (Table 20) for each relation which can be used to generate cloze sentences.

- Google-RE (Petroni et al., 2019) is a subset of knowledge graphs containing three relations: *place of birth*, *date of birth*, and *place of death*. The fact triplets associated with each relation are extracted from Wikipedia and aligned with a short piece of support text. Table 19 shows the predefined template for each relation in Google-RE.

- Wikipedia Biography Dataset (WikiBio) (Sung et al., 2021) is a biology knowledge graph that is constructed by extracting the biology-related facts from Wikidata. Table 21 shows the template for each relation in WikiBio.

- Unified Language Medical System (ULMS) (Bodenreider, 2004) is a medical knowledge graph constructed by domain experts. It contains information about various medical concepts and their relationships. Table 22 shows the template for each relation in UMLS.

Table 3 reports the domains and data statistics.

## A.2  Implementations

**Large Language Model**  We use the HuggingFace implementation of LLaMA and the T5 family. The inference process is run on a single RTX8000 GPU with 48GB memory with Mixed-precision (FP16).

| LLM | #params | Model Implementation |
|-----|---------|---------------------:|
| ChatGPT | - | GPT-3.5-turbo |
| LLaMA-7B | 7B | Touvron et al. (2023) |
| Alpaca | 7B | Taori et al. (2023) |
| Vicuna | 7B | Chiang et al. (2023) |
| T5-XL | 3B | t5-3b |
| FLAN-T5-XL | 3B | google/flan-t5-xl |
| FLAN-Alpaca | 3B | declare-lab/flan-alpaca-xl |
| FLAN-Vicuna | 3B | lmsys/fastchat-t5-3b-v1.0 |

Table 4: Large language model (LLM) description and statistics.

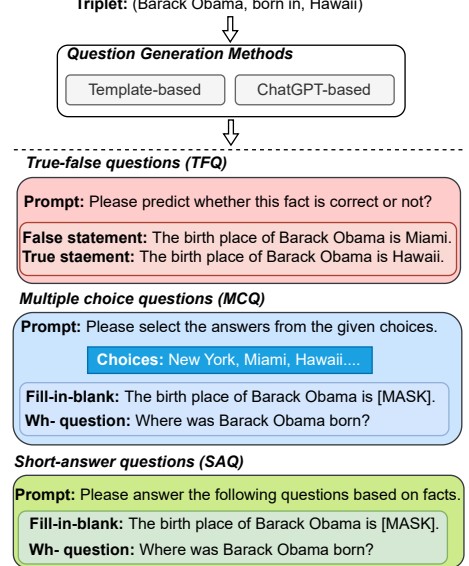

Figure 3: Our question generation process iterates through all fact triplets and creates multiple question types for each triplet.

**Question Generation**  Given a triplet, we generate the TFQ with the ratio of true and false questions set to $1:1$. For MCQ, we randomly select three incorrect entities and combine them with the correct entities as the choices. Table 5 shows the number of generated questions for each KG. We also illustrate the example of template-based and LLM-based questions in Figure 3.

**Abstained Answer Detection**  Assessing the accuracy of answers generated by LLMs in free text format presents a challenge in determining both the correctness of the answer and whether the model chooses to abstain from answering. For TFQ, instead of treating it as a binary classification problem, we instruct the model to respond with "UNKNOWN" when uncertain, effectively transforming it into a 3-class text classification task. For MCQ and ASQ, we compile a curated list of phrases that indicate abstention, such as "cannot

| Question | Google-RE | TREx | WikiBio | UMLS |
|---|---|---|---|---|
| TFQ | 11,056 | 68,078 | 35,164 | 128,610 |
| MCQ | 5,528 | 34,039 | 17,582 | 64,305 |
| SAQ | 5,506 | 32,454 | 7,391 | 35,958 |

Table 5: Number of generated questions for each question type: true-false question (TFQ), multi-choice question (MCQ), short-answer question (SAQ).

| | G_RE | TREx | WikiBio | UMLS |
|---|---|---|---|---|
| TFQ | 90.79 | 92.14 | 90.75 | 90.07 |
| FiB | 92.11 | 92.78 | 91.86 | 89.39 |

Table 6: Similarity (BERT score) between template-based and LLM-based questions, w.r.t two question formats: true/false question (TFQ) and fill-in-blank (FiB).

predict" or "I am sorry," and check if any of these phrases appear in the output. If such phrases are detected, we consider the answer to be an abstained.

**Answer Prompting** Each relation in Google-RE KG comes with the corresponding paragraphs from which it is extracted. We treat this paragraph as relevant context for a given triplet. We sample the paragraph from an unrelated triplet, i.e. not sharing subjects or objects as irrelevant context. For the antifactual context, we replace the correct answer with a randomly selected entity from KG.

## B Additional Results

**Question Analysis** We first evaluate the validity of the LLM-based questions by calculating the similarity between the template-based questions in Table 6. Then, we report the diversity of LLM-based questions in Table 7. Since the templates are written by humans, higher similarities indicate the higher validity of the LLM-based question. From the results in Table 6, we can find that LLM-based questions are highly similar to template-based questions across all datasets w.r.t two question formats: true/false question (TFQ) and fill-in-blank (FiB). This verifies the good quality of the LLM-based questions which can be further used to assess the factual knowledge of LLMs.

Although the accountability of the template, the task of defining diverse templates can be quite burdensome. Due to the lack of templates for Wh-questions, we evaluate the diversity of Wh- questions generated by ChatGPT using the self-bleu scores (Zhu et al., 2018). The lower the scores, the more diverse the questions. From the results

| | | G_RE | TREx | WikiBio | UMLS |
|---|---|---|---|---|---|
| TFQ | TPL | 95.18 | 98.37 | 99.97 | 99.70 |
| FiB | TPL | 74.12 | 79.33 | 87.53 | 87.53 |
| Wh- | GPT3.5 | 50.39 | 64.97 | 81.78 | 78.98 |

Table 7: Diversity measure (self-bleu score) of template-based and LLM-based questions.

in Table 7, we can see that compared to the TFQ and FiB questions generated based on templates. The Wh- questions generated by ChatGPT achieve a higher diversity, which provides a more natural and clear instruction to assess the knowledge.

**F1 score** F1 score on different question types are shown in Tables 8 to 11.

**Precision** Precision score on different KGs are shown in Tables 12 to 14. Similar to Google-RE, ChatGPT are the top performer across all KG, followed by FLAN-T5-XL.

**Recall** Recall score on different KGs are shown in Tables 15 to 18.

**Adversarial Context** The F1 score of different LLMs under different types of context is shown in Figure 4.

## C Example Prompts

**Question Generation Prompt** The question generation prompts for TFQ, FiB and Wh- question can be found in Table 23, Table 24 and Table 25 respectively.

**Answer Prompts** Table 26 provides the prompt template for different LLM families. LLaMA-7B and models finefuned on Alpaca dataset are prompted with the same instruction format. On the other hand, Vicuna and FLAN-T5-XL employ different templates. The instructions also vary for different question types and formats, as shown in Table 27.

| Model | TFQ | | MCQ | | | SAQ | | | AVG |
|---|---|---|---|---|---|---|---|---|---|
| | TPL | GPT3.5 | FiB-TPL | FiB-GPT3.5 | Wh-GPT3.5 | FiB-TPL | FiB-GPT3.5 | Wh-GPT3.5 | |
| ChatGPT | **65.06** | **67.23** | 47.25 | **53.20** | 46.36 | 2.22 | 1.21 | 3.63 | **35.77** |
| LLaMA-7B | 1.11 | 1.10 | 7.20 | 0.76 | 0.27 | 2.92 | 3.07 | 0.15 | 2.07 |
| Alpaca | 60.99 | 53.66 | 41.95 | 40.50 | 41.68 | 7.30 | 6.01 | 8.39 | 32.56 |
| Vicuna | 51.51 | 50.57 | 15.17 | 18.09 | 33.53 | 3.34 | 4.77 | 5.90 | 22.86 |
| T5-XL | 23.85 | 6.76 | 5.77 | 3.96 | 8.23 | 1.63 | 1.69 | 1.33 | 6.65 |
| FLAN-T5-XL | 31.82 | 26.15 | **51.59** | 50.72 | **51.66** | **11.57** | **11.42** | 9.99 | 30.62 |
| FLAN-Alpaca | 54.82 | 53.06 | 46.59 | 47.58 | 48.59 | 9.47 | 8.06 | **10.93** | 34.89 |
| FLAN-Vicuna | 52.92 | 49.91 | 46.80 | 46.76 | 48.67 | 2.74 | 2.89 | 10.83 | 32.69 |

Table 8: F1 on different question types generated from Google_RE KGs.

| Model | TFQ | | MCQ | | | SAQ | | | AVG |
|---|---|---|---|---|---|---|---|---|---|
| | TPL | GPT3.5 | FiB-TPL | FiB-GPT3.5 | Wh-GPT3.5 | FiB-TPL | FiB-GPT3.5 | Wh-GPT3.5 | |
| ChatGPT | **85.43** | **82.38** | **90.32** | **87.17** | **87.61** | **49.17** | **52.57** | **57.38** | **74.00** |
| LLaMA-7B | 2.10 | 3.63 | 2.69 | 3.82 | 4.36 | 31.73 | 19.42 | 0.19 | 8.49 |
| Alpaca | 66.95 | 67.77 | 67.05 | 65.77 | 73.50 | 45.36 | 44.32 | 57.27 | 61.00 |
| Vicuna | 56.52 | 55.87 | 29.62 | 27.46 | 46.74 | 34.54 | 29.27 | 48.58 | 41.08 |
| T5-XL | 20.79 | 21.99 | 7.23 | 4.99 | 12.75 | 6.70 | 1.52 | 14.47 | 11.31 |
| FLAN-T5-XL | 66.88 | 58.98 | 77.46 | 73.80 | 78.87 | 36.14 | 26.78 | 38.22 | 57.14 |
| FLAN-Alpaca | 69.75 | 62.45 | 72.71 | 70.44 | 75.39 | 37.33 | 30.96 | 48.22 | 58.41 |
| FLAN-Vicuna | 70.43 | 64.65 | 74.71 | 70.75 | 76.42 | 19.41 | 18.42 | 41.99 | 54.60 |

Table 9: F1 on different question types generated from TREx KGs.

| Model | TFQ | | MCQ | | | SAQ | | | AVG |
|---|---|---|---|---|---|---|---|---|---|
| | TPL | GPT3.5 | FiB-TPL | FiB-GPT3.5 | Wh-GPT3.5 | FiB-TPL | FiB-GPT3.5 | Wh-GPT3.5 | |
| ChatGPT | **71.27** | **69.07** | **81.77** | **86.57** | **79.07** | **36.45** | **38.72** | **39.02** | **62.74** |
| LLaMA-7B | 0.32 | 3.49 | 0.36 | 0.38 | 2.78 | 1.27 | 1.43 | 0.08 | 1.26 |
| Alpaca | 59.28 | 58.75 | 38.77 | 57.24 | 45.35 | 7.41 | 34.74 | 34.40 | 41.99 |
| Vicuna | 51.15 | 52.55 | 13.02 | 15.45 | 13.54 | 4.42 | 21.20 | 34.90 | 25.78 |
| T5-XL | 16.85 | 29.05 | 5.65 | 6.46 | 5.17 | 0.96 | 9.07 | 2.87 | 9.51 |
| FLAN-T5-XL | 35.43 | 50.36 | 44.23 | 63.17 | 52.57 | 10.44 | 14.09 | 16.30 | 35.82 |
| FLAN-Alpaca | 49.75 | 51.73 | 25.44 | 58.68 | 44.46 | 15.15 | 22.16 | 21.67 | 36.13 |
| FLAN-Vicuna | 55.21 | 54.79 | 45.89 | 63.06 | 49.72 | 3.62 | 3.02 | 17.45 | 36.60 |

Table 10: F1 on different question types generated from WikiBio KGs.

| Model | TFQ | | MCQ | | | SAQ | | | AVG |
|---|---|---|---|---|---|---|---|---|---|
| | TPL | GPT3.5 | FiB-TPL | FiB-GPT3.5 | Wh-GPT3.5 | FiB-TPL | FiB-GPT3.5 | Wh-GPT3.5 | |
| ChatGPT | **68.12** | **54.67** | **70.78** | **83.52** | **63.11** | 21.43 | 18.05 | **12.21** | **48.99** |
| LLaMA-7B | 0.84 | 1.73 | 0.47 | 0.30 | 1.28 | 3.04 | 3.12 | 0.00 | 1.35 |
| Alpaca | 55.21 | 49.60 | 33.66 | 63.32 | 35.87 | 12.71 | **36.55** | 7.80 | 36.84 |
| Vicuna | 51.36 | 50.00 | 12.16 | 21.06 | 11.72 | 7.79 | 19.56 | 10.26 | 22.99 |
| T5-XL | 32.29 | 30.81 | 5.87 | 9.30 | 6.95 | 1.57 | 34.34 | 1.69 | 15.35 |
| FLAN-T5-XL | 23.80 | 34.12 | 41.41 | 69.90 | 45.69 | 12.45 | 8.89 | 6.40 | 30.33 |
| FLAN-Alpaca | 50.81 | 50.31 | 34.63 | 63.48 | 42.47 | 8.91 | 26.70 | 5.82 | 35.39 |
| FLAN-Vicuna | 51.54 | 51.51 | 46.46 | 72.98 | 45.99 | 3.73 | 1.50 | 5.53 | 34.91 |

Table 11: F1 on different question types generated from UMLS KG.

| Model | TFQ | | MCQ | | | ASQ | | | |
|---|---|---|---|---|---|---|---|---|---|
| | TPL | GPT3.5 | FiB-TPL | FiB-GPT3.5 | Wh-GPT3.5 | FiB-TPL | FiB-GPT3.5 | Wh-GPT3.5 | AVG |
| ChatGPT | **88.63** | **85.10** | **91.40** | **88.13** | **88.11** | **57.38** | **58.36** | **64.12** | **77.65** |
| LLaMA-7B | 3.22 | 5.17 | 2.69 | 3.82 | 4.36 | 32.21 | 19.43 | 0.19 | 8.89 |
| Alpaca | 71.89 | 71.52 | 67.05 | 65.80 | 73.50 | 45.75 | 44.39 | 57.31 | 62.15 |
| Vicuna | 57.85 | 57.18 | 29.83 | 27.71 | 46.97 | 34.77 | 29.48 | 51.02 | 41.85 |
| T5-XL | 20.81 | 22.24 | 7.23 | 4.99 | 12.75 | 6.70 | 1.53 | 14.47 | 11.34 |
| Flan-T5-XL | 85.89 | 79.69 | 77.46 | 73.80 | 78.87 | 36.64 | 27.49 | 38.22 | 62.26 |
| Flan-Alpaca | 70.30 | 63.18 | 72.71 | 70.47 | 75.39 | 37.33 | 30.98 | 48.22 | 58.57 |
| Flan-Vicuna | 77.45 | 69.81 | 74.71 | 70.76 | 76.42 | 19.68 | 18.43 | 42.01 | 56.16 |

% invalid responses    0%    25%    50%    75%    100%

Table 12: Precision on different question types in TREx KGs. The shade of background color shows the percentage of invalid responses.

| Model | TFQ | | MCQ | | | SAQ | | | |
|---|---|---|---|---|---|---|---|---|---|
| | TPL | GPT3.5 | FiB-TPL | FiB-GPT3.5 | Wh-GPT3.5 | FiB-TPL | FiB-GPT3.5 | Wh-GPT3.5 | AVG |
| ChatGPT | **75.75** | **72.29** | **81.83** | **86.65** | **79.09** | **39.54** | **40.4** | **41.27** | **64.60** |
| LLaMA-7B | 0.34 | 4.03 | 0.36 | 0.38 | 2.78 | 1.49 | 1.43 | 0.08 | 1.36 |
| Alpaca | 62.08 | 60.33 | 38.77 | 57.24 | 45.35 | 7.94 | 34.74 | 34.58 | 42.63 |
| Vicuna | 52.5 | 53.72 | 13.12 | 15.57 | 13.57 | 4.5 | 21.26 | 35.44 | 26.21 |
| T5 | 17.13 | 29.2 | 5.65 | 6.46 | 5.17 | 0.96 | 9.07 | 2.87 | 9.56 |
| FLAN-T5 | 74.01 | 61.07 | 44.23 | 63.17 | 52.57 | 10.44 | 14.61 | 16.3 | 42.05 |
| FLAN-Alpaca | 52.11 | 51.91 | 25.44 | 58.68 | 44.47 | 15.15 | 22.16 | 21.67 | 36.45 |
| FLAN-Vicuna | 59.65 | 55.38 | 45.89 | 63.06 | 49.72 | 5.18 | 3.91 | 17.45 | 37.53 |

% abstention    0%    25%    50%    75%    100%

Table 13: Precision on different question types generated from wikibio KG. The shade of background color shows the percentage of abstained responses.

| Model | TFQ | | MCQ | | | SAQ | | | |
|---|---|---|---|---|---|---|---|---|---|
| | TPL | GPT3.5 | FiB-TPL | FiB-GPT3.5 | Wh-GPT3.5 | FiB-TPL | FiB-GPT3.5 | Wh-GPT3.5 | AVG |
| ChatGPT | **73.20** | **59.09** | **71.09** | **83.60** | **64.04** | **22.76** | 18.43 | **14.39** | **50.83** |
| LLaMA-7B | 0.94 | 1.96 | 0.47 | 0.30 | 1.28 | 3.04 | 3.12 | 0.00 | 1.39 |
| Alpaca | 57.71 | 53.54 | 33.67 | 63.34 | 35.88 | 12.72 | **36.58** | 7.86 | 37.66 |
| Vicuna | 52.52 | 51.07 | 12.29 | 21.23 | 11.78 | 7.84 | 19.62 | 10.51 | 23.36 |
| T5-XL | 32.32 | 31.40 | 5.87 | 9.31 | 6.95 | 1.57 | 34.36 | 1.69 | 15.43 |
| Flan-T5-XL | 69.66 | 55.00 | 41.44 | 69.94 | 45.73 | 12.45 | 8.89 | 6.40 | 38.69 |
| Flan-Alpaca | 54.63 | 51.56 | 34.67 | 63.53 | 42.52 | 8.91 | 26.72 | 5.82 | 36.05 |
| Flan-Vicuna | 62.52 | 53.10 | 46.49 | 73.01 | 46.02 | 3.73 | 1.50 | 5.53 | 36.49 |

% abstention    0%    25%    50%    75%    100%

Table 14: Precision on different question types generated from UMLS KG. The shade of background color shows the percentage of abstained responses.

| Model | TFQ | | MCQ | | | SAQ | | | AVG |
|---|---|---|---|---|---|---|---|---|---|
| | TPL | GPT3.5 | FiB-TPL | FiB-GPT3.5 | Wh-GPT3.5 | FiB-TPL | FiB-GPT3.5 | Wh-GPT3.5 | |
| ChatGPT | 56.33 | 59.41 | 38.59 | 45.60 | 42.75 | 1.42 | 0.80 | 2.00 | 30.86 |
| LLaMA-7B | 1.01 | 0.89 | 7.20 | 0.76 | 0.27 | 2.92 | 3.07 | 0.15 | 2.03 |
| Alpaca | 57.38 | 48.11 | 41.95 | 40.50 | 41.68 | 6.96 | 6.01 | 8.35 | 31.37 |
| Vicuna | 50.26 | 49.36 | 14.87 | 17.91 | 33.21 | 2.98 | 4.74 | 4.74 | 22.26 |
| T5-XL | 23.83 | 6.74 | 5.77 | 3.96 | 8.23 | 1.63 | 1.69 | 1.33 | 6.65 |
| FLAN-T5-XL | 20.20 | 15.64 | 51.59 | 50.72 | 51.66 | 11.57 | 11.42 | 9.99 | 27.85 |
| FLAN-Alpaca | 54.82 | 53.04 | 46.58 | 47.58 | 48.59 | 9.08 | 8.06 | 10.93 | 34.84 |
| FLAN-Vicuna | 45.24 | 40.99 | 46.80 | 46.76 | 48.64 | 2.74 | 2.89 | 10.82 | 30.61 |

Table 15: Recall on different question types generated from Google_RE KGs.

| Model | TFQ | | MCQ | | | SAQ | | | AVG |
|---|---|---|---|---|---|---|---|---|---|
| | TPL | GPT3.5 | FiB-TPL | FiB-GPT3.5 | Wh-GPT3.5 | FiB-TPL | FiB-GPT3.5 | Wh-GPT3.5 | |
| ChatGPT | 82.46 | 79.84 | 89.27 | 86.23 | 87.12 | 43.02 | 47.83 | 51.93 | 70.96 |
| LLaMA-7B | 1.56 | 2.8 | 2.69 | 3.82 | 4.36 | 31.27 | 19.4 | 0.19 | 8.26 |
| Alpaca | 62.63 | 64.4 | 67.04 | 65.74 | 73.5 | 44.97 | 44.25 | 57.23 | 59.97 |
| Vicuna | 55.25 | 54.61 | 29.42 | 27.21 | 46.5 | 34.31 | 29.06 | 46.37 | 40.34 |
| T5-XL | 20.78 | 21.76 | 7.23 | 4.99 | 12.75 | 6.7 | 1.52 | 14.47 | 11.28 |
| FLAN-T5-XL | 54.76 | 46.81 | 77.46 | 73.8 | 78.87 | 35.66 | 26.1 | 38.22 | 53.96 |
| FLAN-Alpaca | 69.2 | 61.74 | 72.71 | 70.4 | 75.39 | 37.33 | 30.93 | 48.22 | 58.24 |
| FLAN-Vicuna | 64.58 | 60.2 | 74.71 | 70.74 | 76.42 | 19.14 | 18.42 | 41.97 | 53.27 |

Table 16: Recall on different question types generated from TREx KGs.

| Model | TFQ | | MCQ | | | SAQ | | | AVG |
|---|---|---|---|---|---|---|---|---|---|
| | TPL | GPT3.5 | FiB-TPL | FiB-GPT3.5 | Wh-GPT3.5 | FiB-TPL | FiB-GPT3.5 | Wh-GPT3.5 | |
| ChatGPT | 67.28 | 66.13 | 81.72 | 86.49 | 79.04 | 33.82 | 37.18 | 37.00 | 61.08 |
| LLaMA-7B | 0.30 | 3.07 | 0.36 | 0.38 | 2.78 | 1.11 | 1.43 | 0.08 | 1.19 |
| Alpaca | 56.72 | 57.25 | 38.77 | 57.24 | 45.35 | 6.95 | 34.74 | 34.21 | 41.40 |
| Vicuna | 49.88 | 51.42 | 12.92 | 15.33 | 13.51 | 4.34 | 21.15 | 34.39 | 25.37 |
| T5-XL | 28.89 | 5.65 | 6.46 | 5.17 | 0.96 | 9.07 | 2.87 | 9.46 | |
| FLAN-T5-XL | 23.29 | 42.84 | 44.23 | 63.17 | 52.57 | 10.44 | 13.61 | 16.30 | 33.31 |
| FLAN-Alpaca | 47.59 | 51.56 | 25.44 | 58.68 | 44.45 | 15.15 | 22.16 | 21.67 | 35.84 |
| FLAN-Vicuna | 51.38 | 54.21 | 45.89 | 63.06 | 49.72 | 2.78 | 2.46 | 17.45 | 35.87 |

Table 17: Recall on different question types generated from wikibio KGs.

| Model | TFQ | | MCQ | | | SAQ | | | AVG |
|---|---|---|---|---|---|---|---|---|---|
| | TPL | GPT3.5 | FiB-TPL | FiB-GPT3.5 | Wh-GPT3.5 | FiB-TPL | FiB-GPT3.5 | Wh-GPT3.5 | |
| ChatGPT | 63.70 | 50.86 | 70.48 | 83.43 | 62.21 | 20.24 | 17.70 | 10.60 | 47.40 |
| LLaMA-7B | 0.75 | 1.55 | 0.47 | 0.30 | 1.28 | 3.04 | 3.12 | 0.00 | 24.36 |
| Alpaca | 52.91 | 46.19 | 33.65 | 63.30 | 35.85 | 12.71 | 36.52 | 7.74 | 18.71 |
| Vicuna | 50.24 | 48.98 | 12.04 | 20.90 | 11.37 | 7.75 | 19.50 | 10.02 | 29.35 |
| T5-XL | 29.29 | 28.36 | 5.86 | 9.29 | 6.94 | 1.57 | 34.31 | 1.69 | 18.63 |
| FLAN-T5-XL | 14.35 | 24.73 | 41.38 | 69.87 | 45.65 | 12.45 | 8.89 | 6.40 | 30.13 |
| FLAN-Alpaca | 47.49 | 49.11 | 34.59 | 63.43 | 42.43 | 8.91 | 26.67 | 5.82 | 31.39 |
| FLAN-Vicuna | 43.84 | 50.00 | 46.44 | 72.94 | 45.95 | 3.73 | 1.50 | 5.53 | 34.27 |

Table 18: Recall on different question types generated from UMLS KGs.

| Relation | Type | Template |
|---|---|---|
| date_of_birth | N-1 | The birth date of [X] is [Y]. |
| place_of_birth | N-1 | The birth place of [X] is [Y]. |
| place_of_death | N-1 | The death place of [X] is [Y]. |

Table 19: Examples of question generation template for Google_RE, where [X] denotes the subject, and [Y] denotes the object.

| Relation | Type | Template |
|---|---|---|
| capital | 1-1 | The capital of [X] is [Y]. |
| member of political party | N-1 | [X] is a member of the [Y] political party. |
| shares border with | N-M | [X] shares border with [Y]. |

Table 20: Examples of question generation template for Trex, where [X] denotes the subject, and [Y] denotes the object.

| Relation | Type | Template |
|---|---|---|
| drug used for treatment | N-M | The standard treatment for patients with [X] is a drug such as [Y]. |
| medical condition treated | N-M | [X] has effects on diseases such as [Y]. |
| therapeutic area | N-M | [X] cures diseases such as [Y]. |

Table 21: Examples of question generation template for WikiBio, where [X] denotes the subject, and [Y] denotes the object.

| Relation | Type | Template |
|---|---|---|
| may_be_prevented_by | N-M | [X] treats [Y]. |
| gene_mapped_to_disease | N-M | [X] has a genetic association with [Y]. |
| may_be_finding_of_disease | N-M | [X] has symptoms such as [Y]. |

Table 22: Examples of question generation template for UMLS, where [X] denotes the subject, and [Y] denotes the object.

---

**TRUE-FALSE QUESTION**

I have a triplet extracted from a knowledge graph. The triplet is organized as (Subject, Relation, Object), which describes the relation between object and relation. Can you help me to generate a natural language sentence to describe this triplet as accurate as possible?

*{ triplet }*

---

Table 23: Question generation prompts for true-false question format.

---

**FILL-IN-BLANK QUESTION**

I have a triplet extracted from a knowledge graph. The triplet is organized as (Subject, Relation, Object), which describes the relation between object and relation. Can you help me to generate a natural language sentence to describe this triplet as accurate as possible and replace Object with [MASK]?

*{ triplet }*

---

Table 24: Question generation prompt for fill-in-blank question format.

| WH- QUESTION |
| --- |
| I have a triplet extracted from a knowledge graph. The triplet is organized as (Subject, Relation, Object), which describes the relation between object and relation. Can you help me to generate a question based on this triplet that the object is the corresponding answer? Please return the question only. |
| *{ triplet }* |

Table 25: Question generation prompt for Wh- question format.

| | |
| --- | --- |
| `ChatGPT` | *{ context }*
*{ intructions }*
*{ question }* |
| `LLaMA-7B, Alpaca,`
`FLAN-Alpaca` | Below is an instruction that describes a task, paired with an input that provides further context. Write a response that appropriately completes the request.

### Instruction:
*{ context }*
*{ intructions }*
### Input:

*{ question }*
### Response: |
| `Vicuna,`
`FLAN-Vicuna` | *{ context }*
*{ intructions }*

HUMAN:
*{ question }*

ASSISTANT: |
| `T5-XL,Flan-T5-XL` | *{ context }*
*{ intructions }*

QUESTION: *{ question }* |

Table 26: Inference prompt format for different LLMs.

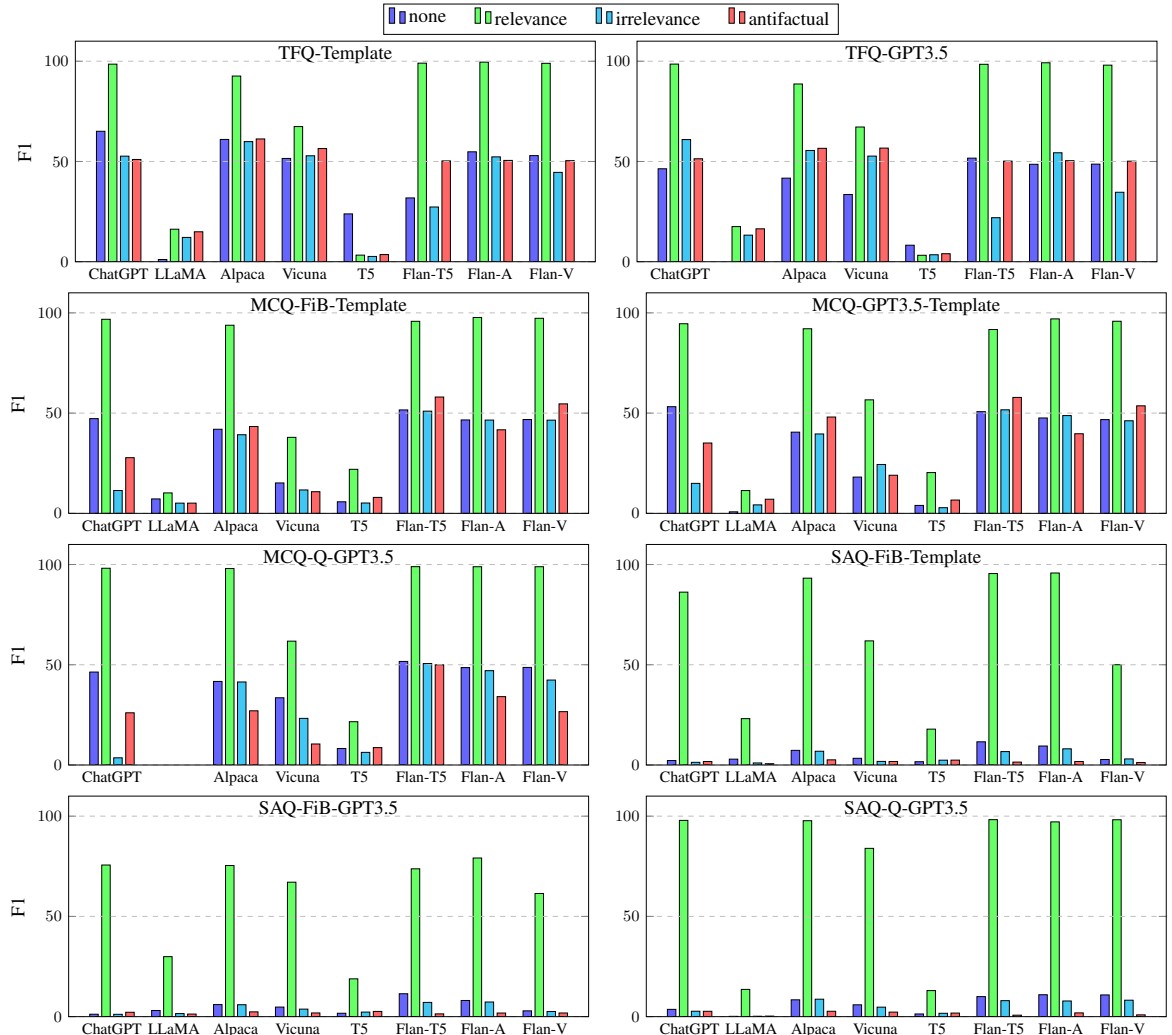

Figure 4: F1 score of LLM on google_re with different context prompt: no context (none), relevant, irrelevant and antifactual context.

| TFQ | The following sentence describes a real-world fact. Please predict whether this fact is correct or not? Please only return correct or incorrect. If don't know, please answer UNKNOWN. If correct, answer CORRECT. If incorrect, answer INCORRECT. |
|-----|------------------------------------------------------------------------------------------------------------------------------------------------------------------------------------------------------------------------------------------------------|
| FiB | Please predict the missing words to complete the following sentence based on the facts in the real-world. The missing words are represented by [MASK]. Please return the missing words only. |
| MCQ | Please answer the following questions based on the facts in the real-world. Please select the answers from the given choices and return the answer only.
Choices: { *choices* } |
| SAQ | Please answer the following questions based on the facts in the real-world. Please keep the answer as simple as possible and return the possible answers as a list. |

Table 27: Instructions for different question type.