# OpenReview forum: "Systematic Assessment of Factual Knowledge in Large Language Models"
_EMNLP/2023/Conference — EMNLP 2023 Findings_

### Official Review · Reviewer_ezC3 · 2023-08-03

**Soundness:** 3

**Excitement:**

3: Ambivalent: It has merits (e.g., it reports state-of-the-art results, the idea is nice), but there are key weaknesses (e.g., it describes incremental work), and it can significantly benefit from another round of revision. However, I won't object to accepting it if my co-reviewers champion it.

**Paper Topic And Main Contributions:**

The paper proposes a method for evaluating the factual knowledge in LLMs. The authors describe a method for creating a collections of questions, in different formats,  from a KB and apply it to several KBs to evaluate several LLMs.

The main contributions of this paper are:

- the method for creating questions from KBs
- the use of different format of questions for the same facts
- evaluation of different families of LLMs
- evaluation of different inputs to the LLMs
- measure abstention

**Questions For The Authors:**

A: line 119, how do you assure the entity is wrong? do you use entities of the same type or different type? It is important to give more details about these wrong entities.
B: line 127, "incorrect entities": the same questions given above

**Reasons To Accept:**

- The paper is clear and contains lot of experiments and information for evaluating LLMs.
- The authors evaluate several LLMs
- The authors use several datasets


**Reasons To Reject:**

- I miss a deeper motivation for assessing factual knowledge instead other types of knowledge. After referring in the introduction to other approaches focused on assessing language and knowledge capabilities, you should explain why you proposed to assess factual knowledge instead. I mean, why it is important to assess factual knowledge in LLMs besides, or beyond, other types of knowledge?
- There is no related work.
- Line 224, "Overall, ChatGPT achieves the best average precision. However, it also has a high percentage of abstained answers across all question types."-> For other systems, besides ChatGPT, the more abstained answers, the better precision. Authors should analyze, and comments, this and include also recall scores in the paper (I only see precision and F1)


**Reproducibility:**

4: Could mostly reproduce the results, but there may be some variation because of sample variance or minor variations in their interpretation of the protocol or method.

**Reviewer Confidence:**

5: Positive that my evaluation is correct. I read the paper very carefully and I am very familiar with related work.

**Typos Grammar Style And Presentation Improvements:**

- line 23, "large language models (LLMs)" -> "Large Language Models (LLMs)". There are several examples across the paper  (NLP, KGs) ... You must capitalized the first letter of each word.

- Line 70, "LLMs performance depends on several factors such as instruction finetuning, domains, and question complexity. Despite sharing the same parametric knowledge base, models finetuned with different instruction datasets show varying performance levels. In general076
domain KGs, LLMs achieve the highest score, but their performance declines in specialized domains and is worse on questions having a wide range of potential answers." -> this is an observation or conclusion from your work, not a contribution. The contribution might be to obtain such conclusion. Something like "We show how LLMs performance ..."

---

> ### Author Rebuttal · Authors · 2023-08-29
>
> Thank you for reviewing our paper and providing insightful feedback. We have carefully considered your points and prepared responses to address your concerns.
>
> > **why it is important to assess factual knowledge in LLMs besides, or beyond, other types of knowledge?**
>
> Assessing factual knowledge in Large Language Models (LLMs) such as ChatGPT is important for several reasons.
> - Factual knowledge refers to information that is objectively true and can be verified against empirical evidence, which is crucial in various downstream tasks, e.g., question answering.
> - Evaluating the factual knowledge of LLMs can ensure that the model is providing reliable and trustworthy information to users. This is crucial, especially in contexts where misinformation or false information can have real-world consequences.
> - Assessing the factual knowledge can provide us with a general understanding of the ability of LLMs before we put them into to actual applications. For example, an LLM without knowledge of the medical domain cannot be applied to the medical diagnosis task.
> - Factual knowledge is a general and fundamental type of knowledge. Other types of knowledge, e.g., commonsense knowledge and reasoning knowledge, can be grouped under factual knowledge or relied on. Thus, assessing factual knowledge is a fundamental task in evaluating LLMs. In the future, we will also explore evaluating other knowledge in LLMs.
>
> > **There is no related work.**
>
> Due to the page constraint, we briefly mentioned the related works on evaluating LLMs in the introduction. We will add a related work section in camera ready (if accepted) when we have an additional page.
>
> > **Authors should analyze, and comments, this and include also recall scores in the paper**
>
> The abstention rate is highlighted in color cell of Table 1. We will improve our presentation and add recall scores in the camera ready (if accepted).
>
> >  **True-False Question, how do you assure the entity is wrong? do you use entities of the same type or different type?**
>
> The KGs studied in our paper do not have type attributes associated with the entities, i.e. there is only a single entity type for each relation.  We first gather all the entities from triples that fall under a specific relation to form a candidate set. This helps us group entities with similar characteristics. To generate false questions, we randomly choose entities from the candidate set that are not linked to the subject under this relation as the incorrect object.

---

### Official Review · Reviewer_yo9m · 2023-08-04

**Soundness:** 2

**Excitement:**

2: Mediocre: This paper makes marginal contributions (vs non-contemporaneous work), so I would rather not see it in the conference.

**Missing References:**

Please refer to reason to reject.

**Paper Topic And Main Contributions:**

This paper proposes a framework to evaluate the factual knowledge in LLMs. They first generate three kinds questions from tuples in knowledge graph, and evaluate the accuracy of different LLMs in answering these questions. By comparing the performance of several LLMs, and they find that ChatGPT outperforms other LLMs, and the performance depends on instruction fine-tuning, domain, context, etc.

**Questions For The Authors:**

Please refer to reason to reject.

**Reasons To Accept:**

* Due to its relation to the issue of model hallucination, the factual knowledge in large-scale models is a matter of great concern. The accurate and unbiased evaluation of factual knowledge within language models is an important research direction.
* This work provides relevant questions and metrics for evaluating the knowledge of language models. It also compares different models, which contributes to a better understanding of Large Language Models and will be beneficial for future evaluations of other models.

**Reasons To Reject:**

* My major concern is whether the benchmark proposed in the paper can genuinely assess factual knowledge in language models. In my opinion, Alpaca and Vicuna are just fine-tuned versions of llama on instruction datasets, which do not involve a significant amount of factual knowledge. Therefore, the factual knowledge in Alpaca and Vicuna should not differ significantly from llama. However, in the paper's experiments, there is a substantial performance gap between these models (34 V.S. 2.1). In other words, the benchmark might be evaluating the models' ability to follow instructions rather than accurately assessing their factual knowledge. Does this also imply that the benchmark is not suitable for evaluating pre-trained language models without instruction tuning?
* About the motivation of this paper:: The author believes that other benchmarks may be affected by the overlap between the model's pre-training data and testing data. However, I don't understand why the evaluation method proposed in this paper would not be affected by such overlap [1].
* About the diversity of questions：The author modified the format of the questions but failed to address another crucial concern, which is the model's sensitivity to prompts/templates [1][2][3]...
* Regarding the novelty of the conclusions, lots of previous works have already discussed the influence of context information on model predictions. The author should incorporate a discussion of these works[4][5][6]....

[1] Can Prompt Probe Pre-trained Language Models? Understanding the Invisible Risks from a Causal View

[2] Measuring and Improving Consistency in Pretrained Language Models

[3] Calibrate Before Use: Improving Few-shot Performance of Language Models

[4] How Context Affects Language Models' Factual Predictions

[5] Rethinking the Role of Demonstrations: What Makes In-Context Learning Work?

[6] A Drop of Ink Makes a Million Think: The Spread of False Information in Large Language Models

**Reproducibility:**

5: Could easily reproduce the results.

**Reviewer Confidence:**

4: Quite sure. I tried to check the important points carefully. It's unlikely, though conceivable, that I missed something that should affect my ratings.

---

> ### Author Rebuttal · Authors · 2023-08-29
>
> Thank you for reviewing our paper and providing insightful feedback. We have carefully considered your points and prepared responses to address your concerns.
>
> > **Whether the benchmark proposed in the paper can genuinely assess factual knowledge in language models.**
>
> In our framework, we focus on evaluating the knowledge of instruction tuning LLMs. Assessing knowledge in instruction tuning LLMs can be done from two perspectives:
> - whether they have such knowledge, usually from the pre-training datasets. Additionally, different instruction-tuning datasets can enhance different skills and may lead to catastrophic forgetting of the knowledge learned in the pretraining stage [1].
> - whether they can accurately express their knowledge,i.e. instruct following ability and robustness
> Given the same factual knowledge, we generate multiple questions at different levels of difficulty, ranging from easy (true-false question) to hard (short answer question) with different contexts. This allows us to rigorously assess whether such knowledge is stored in the LLM.  Alpaca, Vicuna, and LLaMA are pre-trained on the same training datasets, indicating they should have the same factual knowledge. The performance difference indicates the different effect of instruction tuning in learned factual knowledge in the LLaMA backbone. Additionaly, this also show that they have different abilities in expressing their knowledge.
>
> [1] Wang, Y., Ivison, H., Dasigi, P., Hessel, J., Khot, T., Chandu, K. R., ... & Hajishirzi, H. (2023). How Far Can Camels Go? Exploring the State of Instruction Tuning on Open Resources. arXiv preprint arXiv:2306.04751.
>
> > **why the evaluation method proposed in this paper would not be affected by such overlap**
>
> The overlap between public benchmarks and pretraining dataset can hinder the reliability of the evaluation results of the LLMs. Ensuring the non-overlap between training and testing dataset is not the goal of our proposed evaluation framework. We aim to achieve a reliable and rigorous evaluation of factual knowledge stored in LLMs. Instead of relying on an off-the-shelf benchmark, we propose to generate a diverse test set containing different types of questions at varying difficulty levels to reduce the impact of benchmark contamination on the evaluation results.
>
> > **failed to address another crucial concern: the model's sensitivity to prompts/templates**
>
> We would like to point out that our proposed framework is orthogonal to the model prompting techniques. Users have the flexibility to design the prompts to interact with the LLMs.
>
> However, we agree that LLMs tend to be sensitive to the prompts and templates. Therefore, for each LLM, we use the instruction template, which has a similar structure to its instruction finetuning template, to avoid the discrepancy between instruction finetuning and testing. We use the same prompts for all LLMs to ensure the results' consistency. It is indeed an interesting analysis on prompt sensitivity. As it is not our main objective, we leave the exploration of the evaluation prompts as future works.
>
> > **Influence of context information on model predictions. The author should incorporate a discussion of these works**
>
> We thank the reviewer for pointing out the missing references. We will incorporate them in the camera ready (if accepted).

---

### Official Review · Reviewer_HmbN · 2023-08-18

**Soundness:** 3

**Excitement:**

2: Mediocre: This paper makes marginal contributions (vs non-contemporaneous work), so I would rather not see it in the conference.

**Paper Topic And Main Contributions:**

This paper proposes a framework to evaluate factual knowledge in LLMs by generating valid, diverse and widely-covering questions. They leverages facts represented as triplets in Knowledge Graphs to automatically generate: True-False Question, Multiple Choice Question and Short Answer Question. Then use designed prompts to instruct LLMs, while also injecting relevant, irrelevant and anti-factual knowledge in questions to test robustness and consistency.

**Reasons To Accept:**

This paper proposes a method to automatically generate 3 types of questions from knowledge graphs to assess factual knowledge of LLMs, which is an important topic to inspect LLMs on extrinsic hallucinations.

Knowledge graphs representing facts is more easier to obtain than Q&A datasets. This method might saves effort in constructing domain-specific or general datasets used for assessing factual knowledge of models ( or at least provide inspiration?).

Other consideration like the robustness of LLMs and taking abstention into account when evaluating for fairness, makes the work more complete and rigorous.


**Reasons To Reject:**

The idea of automatically generating questions from knowledge graphs must have been utilized somewhere else, making this work less valuable, since it is just ‘moving’ a method into the context of assessing factual knowledge of LLMs. So, I think this work is more like an empirical study.

The difference of this paper and previous works seems less significant. The authors kinda blur the discussion on whether their selected datasets has leakage problems, which is one of their motivations. Besides, another motivation of easier dataset-construction is also not sufficiently proved or demonstrated, making me wonder ‘why on earth use the knowledge graphs...?’

This paper feels more about evaluating LLMs and presenting the evaluation results  but rather the effectiveness and innovativeness of their proposed framework, which makes me confused.


**Reproducibility:**

3: Could reproduce the results with some difficulty. The settings of parameters are underspecified or subjectively determined; the training/evaluation data are not widely available.

**Reviewer Confidence:**

4: Quite sure. I tried to check the important points carefully. It's unlikely, though conceivable, that I missed something that should affect my ratings.

---

> ### Author Rebuttal · Authors · 2023-08-29
>
> Thank you for reviewing our paper and providing insightful feedback. We have carefully considered your points and prepared responses to address your concerns.
>
> > **Automatically generating questions from knowledge graphs must have been utilized somewhere else.**
>
> Although there are previous works that generate questions from knowledge graphs [1], to the best of our knowledge, none of them adopt the generated questions for evaluating the factual knowledge in LLMs.
>
> > **The difference of this paper and previous works seems less significant.**
>
> There are two main motivations for our proposed evaluation frameworks.
> - (i) The lack of benchmark datasets for a specific domain: our framework utilizes KG in the domain of interest to generate evaluation benchmarks. By enumerating through all available factual triplets, we ensure the factual knowledge coverage of the generated benchmarks.
> - (ii) Benchmark data contamination where the test set may appear in the pretraining dataset. Due to the large scale of pretraining data, estimating the overlap ratio between pretraining dataset and test set is time-consuming and requires lots of computational resources. Instead of relying on the off-the-shelf benchmarks, we propose to generate a diverse test set containing different types of questions at varying difficulty levels to ensure rigorous evaluation.
>
> Some prior works attempt to evaluate knowledge in LLMs [2]. However, they lack a systematic framework, including diverse question generation with adequate knowledge coverage and evaluation. Furthermore, their methods may not be suitable for evaluating generative LLMs due to the need to finetune a prediction module. Moreover, they often use the accuracy metric and do not take abstention answers into account, which is essential for a fair evaluation.
>
> > **why on earth use the knowledge graphs...?**
>
> The knowledge graphs contain enormous factual knowledge in a structural format, which is transparent to users. Thus, we try to utilize the KGs to evaluate the knowledge implicitly stored in LLMs. Since the factual knowledge in KGs is structurally represented (subject, relation, object), we can automatically generate the question-answer pair to assess the existence of knowledge in LLMs.
>
>  Moreover, there are many domain-specific KGs available, which can be used to evaluate the knowledge of LLMs in specific areas, such as medical domain. In our experiments, we choose two kinds of KGs: (i) general KGs (Trex and Google-Re) and (ii) domain-specific KGs (Wiki-bio and UMLS) to assess both general factual knowledge and domain-specific knowledge.
>
> > **This paper feels more about evaluating LLMs and presenting the evaluation results but rather the effectiveness and innovativeness of their proposed framework**
>
> In this paper, we provide a systematic framework containing question generation and evaluation modules. This allows users to evaluate the factual knowledge stored in LLMs more effectively, instead of relying on off-the-shelf benchmarks. The experiment results are illustrated to demonstrate the effectiveness of our framework in assessing knowledge in LLMs and present the potential results when using our framework. Our framework could be readily applied to assess the knowledge of LLMs before deployment in high-stake applications such as medical diagnosis.
>
> - [1] Seyler, D., Yahya, M., & Berberich, K. (2017, October). Knowledge questions from knowledge graphs. In Proceedings of the ACM SIGIR International Conference on Theory of Information Retrieval (pp. 11-18).
> - [2] Petroni, F., Rocktäschel, T., Lewis, P., Bakhtin, A., Wu, Y., Miller, A. H., & Riedel, S. (2019). Language models as knowledge bases?. arXiv preprint arXiv:1909.01066.

---

### Official Review · Reviewer_njva · 2023-08-21

**Soundness:** 3

**Excitement:**

3: Ambivalent: It has merits (e.g., it reports state-of-the-art results, the idea is nice), but there are key weaknesses (e.g., it describes incremental work), and it can significantly benefit from another round of revision. However, I won't object to accepting it if my co-reviewers champion it.

**Paper Topic And Main Contributions:**

This paper proposes a framework to systematically evaluate the factual knowledge of LLMs with KGs in generic and specific domains.

**Reasons To Accept:**

**Strengths**
1. LLM evaluation is a timely problem and worth attention.
2. This paper conducts a comprehensive analysis which covers various LLMs, question types and knowledge domains.

**Reasons To Reject:**

**Weaknesses：**
1. The proposed evaluation pipeline is similar to prior works.
2. ChatGPT shows a great percentage of abstention than other models. Is that fair to compare their accuracies?
3. Reproducibility. The author doesn't mention if the code will be available to the public.

**Reproducibility:**

3: Could reproduce the results with some difficulty. The settings of parameters are underspecified or subjectively determined; the training/evaluation data are not widely available.

**Reviewer Confidence:**

4: Quite sure. I tried to check the important points carefully. It's unlikely, though conceivable, that I missed something that should affect my ratings.

---

> ### Author Rebuttal · Authors · 2023-08-29
>
> We would like to express our gratitude for taking the time to review our paper and the insightful feedback you have provided. We have carefully considered each of your points and have prepared responses to address the concerns you raised.
>
> > **The proposed evaluation pipeline is similar to prior works.**
>
> Evaluation of the Large Language Model (LLM) has gained increasing interest among researchers. Despite prior works attempt to evaluate knowledge in LLMs [1], most lacks a systematic framework for rigorous evaluation and often cannot be applied to recent generative LLMs. Our work, on the other hand, proposes a novel framework for evaluating the factual knowledge of arbitrary LLMs such as ChatGPT and Alpaca.  Our proposed framework has the capability to tackle the absence of domain-specific benchmarks as well as mitigate the leakage issue of public off-the-shelf benchmarks for a more reliable assessment.
>
> To do so, we  generate three types of questions from Knowledge Graphs (KGs): True/False Questions (TFQ), Multiple Choice Questions (MCQ), and Short Answer Questions (SAQ). These question types allow us to access different levels of knowledge in LLMs while ensuring comprehensive knowledge coverage, i.e. we generate questions for all factual triplets. Additionally, we introduce a new F1 metric that addresses the abstention problem commonly found in advanced LLMs. This metric is more suitable for evaluating LLM performance.
>
> We appreciate it if the reviewer could additionally guide us on any missing prior works that we may have overlooked.
>
> > **ChatGPT shows a great percentage of abstention than other models. Is that fair to compare their accuracies?**
>
> Many LLMs may return abstention answers to prevent hallucination and misuse. LLMs with higher abstention tend to avoid providing inaccurate or hallucinated answers, which prioritizes the precision metrics, i.e. accuracy. For a fair comparison among different LLMs, we introduce the recall metric in Eq.3, which takes the abstention answer into account. Furthermore, we introduce the F1 score, which balances the precision and recall metrics, providing a general metric for evaluation. The results of F1 of different LLMs can be found in the Appendix.
>
> > **Reproducibility. The author doesn't mention if the code will be available to the public.**
>
> We thank the reviewer for the interest in our framework. We will release both the code of our framework and the generated test sets from each KG upon acceptance.
>
> [1] Petroni, F., Rocktäschel, T., Lewis, P., Bakhtin, A., Wu, Y., Miller, A. H., & Riedel, S. (2019). Language models as knowledge bases?. arXiv preprint arXiv:1909.01066.

---

### Meta-Review · Area_Chair_67NF · 2023-09-17

**Recommendation:** 3

**Metareview:**

The paper proposes a framework to systematically evaluate factual knowledge of LLMs with KGs in generic and specific domains. All reviewers appreciated that the analysis in the paper is comprehensive, covering various LLMs, question types, and domains. Reviewers highlighted how the approach takes into account abstentions by the LLMs.

Reviewers also expressed some concerns about the experiment setup. For instance, it's not clear if the evaluation might be conflating models' ability to follow instructions with factual knowledge. Another issue that reviewers raised is whether the proposed benchmark also has ‘leakage problems’ similar to past benchmarks. One reviewer pointed out that automatically generating questions from knowledge graphs has been explored in past work (see reviews for references) and shouldn’t be considered a major contribution of this paper.

---

### Decision · Program_Chairs · 2023-10-07

**Decision:**

Accept-Findings

**Comment:**

The paper proposes a framework to systematically evaluate factual knowledge of LLMs with KGs in generic and specific domains. All reviewers appreciated that the analysis in the paper is comprehensive, covering various LLMs, question types, and domains. Reviewers highlighted how the approach takes into account abstentions by the LLMs.

Reviewers also expressed some concerns about the experiment setup. For instance, it's not clear if the evaluation might be conflating models' ability to follow instructions with factual knowledge. Another issue that reviewers raised is whether the proposed benchmark also has ‘leakage problems’ similar to past benchmarks. One reviewer pointed out that automatically generating questions from knowledge graphs has been explored in past work (see reviews for references) and shouldn’t be considered a major contribution of this paper.